## THE NATURAL HISTORY OF MODEL ORGANISMS

# The big potential of the small frog *Eleutherodactylus coqui*

**Abstract**   The Puerto Rican coquí frog *Eleutherodactylus coqui* is both a cultural icon and a species with an unusual natural history that has attracted attention from researchers in a number of different fields within biology. Unlike most frogs, the coquí frog skips the tadpole stage, which makes it of interest to developmental biologists. The frog is best known in Puerto Rico for its notoriously loud mating call, which has allowed researchers to study aspects of social behavior such as vocal communication and courtship, while the ability of coquí to colonize new habitats has been used to explore the biology of invasive species. This article reviews existing studies on the natural history of *E. coqui* and discusses opportunities for future research.

**SARAH E WESTRICK\*, MARA LASLO AND EVA K FISCHER**

**\*For correspondence:** westse@illinois.edu

**Competing interest:** The authors declare that no competing interests exist.

## Introduction

On the Caribbean island of Puerto Rico the night is filled with high-pitched calls of "¡Ko-kee!" as the coquí común – the common or Puerto Rican coquí (*Eleutherodactylus coqui*) – sings its song across the island. This small but boisterous frog is a national symbol of Puerto Rico and has featured prominently in the island's culture for thousands of years (*Joglar, 1998*). The indigenous Taíno people believe a goddess created the frog to forever call out the name of her lost love, Coquí (pronounced ko-kee or co-qui), who was taken from her by the god of chaos and disorder. The coquí frog remains as celebrated today as it was by the ancient Taíno people, and the Taíno symbol for the frog (*Figure 1*) appears in everything from artwork and pottery to the marketing and branding many of Puerto Rican companies. Indeed, the coquí frog is so important to the identity of Puerto Ricans that they often express their nationality by saying "Soy de aquí como el coquí" ("I'm from here, like the coquí").

However, in addition to being an important cultural symbol in Puerto Rico, *E. coqui* has also become an important species for scientific inquiry, and researchers from a number of fields have been drawn to the coquí frog as a result of its natural history, conspicuous behavior, and sheer abundance. This combination of traits has made the coquí frog an excellent model for the study of developmental biology, neuroethology (notably in the areas of mate attraction, communication, and auditory processing) and invasion biology. Here, we give an overview of the life history of the coquí frog, discuss its unique role in three major fields of biology, outline research with coquí in other areas (*Box 1*), and highlight open questions about the natural history of the coquí frog and its relatives.

## Life history of the coquí

Common coquí are small, nocturnal, terrestrial frogs (*Thomas, 1965*). The snout-vent length of a coquí is up to 63 mm for adult females and 50 mm for adult males. Coquí are gray brown, with natural variation in color and striping among populations (*Figure 2A–C*; *Beard et al., 2009*; *Woolbright and Stewart, 2008*). This polymorphic variation is driven by local habitat matching and reduces predation risk by apostatic selection, where predators have a search image of the common morph, thus favoring survival of the rarer morphs (*Woolbright and Stewart, 2008*). Coquí have clutches of 10–40 eggs that are fertilized internally and brooded on land (*Elinson et al., 1990*). Males guard eggs as they develop directly from egg to froglet, skipping the free-swimming tadpole stage observed in most frogs (*Townsend, 1986*; *Townsend and Stewart, 1985*). The small size of coquí froglets makes

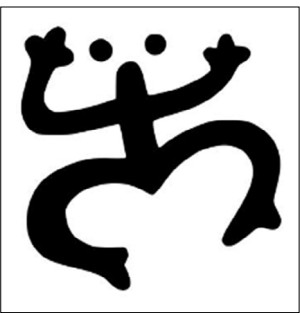

**Figure 1.** The indigenous Taíno symbol for coquí, which is ubiquitous in Puerto Rico. The frog-like hands seen in Taíno imagery are associated with 'femaleness' and the calls of coquí are associated with female fertility and children (***Ostapkowicz, 2015***).

them vulnerable to predation by many invertebrates such as the giant crab spider (*Olios* spp.; ***Formanowicz et al., 1981***).

Coquí are generalists occurring in a wide range of habitats, including forests, mountains, and urban areas in tropical regions of the Americas. Since *E. coqui* are terrestrial and lay eggs on land, they are not constrained by needing bodies of water for reproduction, provided sufficient humidity (***Beard and Pitt, 2012***). Coquí show a shift in microhabitat preference during development. While adults can be found across the whole vertical spectrum, from leaf litter to canopies, juveniles are more often found in the understory and avoid leaf litter (***Beard et al., 2003***). At night, adult coquí prefer large-leafed plants that can support their weight for calling and foraging and will use large fallen leaves for nesting and diurnal retreat sites (***Beard et al., 2003***; ***Townsend, 1989***). Coquí are primarily insectivores and opportunistic feeders of abundant prey (***Beard, 2007***; ***Stewart and Woolbright, 1996***; ***Woolbright and Stewart, 1987***).

In their native Puerto Rico, coquí live at high densities. Estimates vary but can be as high as ~50,000 frogs/hectare (***Beard et al., 2008***; ***Stewart and Woolbright, 1996***; ***Woolbright et al., 2006***). Populations of coquí are estimated to consume up to ~690,000 invertebrates/hectare each night (***Beard et al., 2008***). Population densities are impacted by rainfall, the number of available territories, and disruptive weather events such as hurricanes (***Stewart and Pough, 1983***; ). As habitat generalists living closely with humans, and liberated from water for reproduction thanks to terrestrial breeding, coquí have readily invaded other Caribbean and Pacific islands through accidental introductions (***Figure 3*** and detailed discussion below; ***Beard***

*and Pitt, 2012*; ***Stewart and Woolbright, 1996***). Indeed, a single male brooding a clutch of eggs in a potted plant may be sufficient to establish a new population (***Figure 3*** photo inset).

## Development

The direct-developing life cycle of coquí is dramatically different from the ancestral metamorphosing life cycle observed in most living frogs. Therefore, *E. coqui* is an important model for understanding the evolutionary origins and consequences of direct development. Direct-developing frogs skip the free-swimming tadpole stage characteristic of most frogs, including *Xenopus laevis*, the most common model of amphibian development. Instead, direct-developing frogs hatch as miniature adults (***Figure 2D–H***).

Direct development has evolved independently multiple times in frogs (***Duellman and Trueb, 1994***; ***Gomez-Mestre et al., 2012***; ***Figure 4A***), suggesting that this life history can be advantageous. This is likely because – especially when coupled with parental care as in coquí (see below) – direct-developing frogs are freed from the requirement of water for breeding.

Several immediate questions arise about this unique life history: Is early development and patterning (the process by which equivalent cells take on different identities) the same in direct-developing and metamorphosing frogs? To what extent do direct-developing frogs repeat tadpole development within the egg? In coquí, the answer seems to be that development is a mix of conserved and novel features. This unique combination of developmental features has made coquí an important model for the evolution of development, as well as a model for the evolution of the amniote egg, a key innovation in the evolutionary diversification of vertebrates.

### Early development

While fertilization in most frogs is external, fertilization occurs internally in coquí (***Townsend et al., 1981***; ***Townsend and Stewart, 1985***). Because there is no independently feeding tadpole stage, the egg must contain all the nutrients needed to get the embryo to the juvenile froglet stage. As a result, coquí eggs are large: 3.5 mm as compared to the 1.3 mm eggs of *Xenopus laevis*.

Frogs in general have holoblastic, or complete, cleavage of the egg. Coquí maintain this complete cleavage, but the large egg shifts the cleavage furrows towards the animal pole (top), resulting in a larger vegetal (bottom) section (***Figure 5***).

## Box 1. Other notable coquí research

### Fossil evidence

As one of the most speciose vertebrate genera, Eleutherodactylids have a long evolutionary history. In fact, a distal humerus of an Eleutherodactylid found in Puerto Rico appears to be the earliest known fossil frog from any Caribbean island, with an estimated age of ~29 Mya (Oligocene; *Blackburn et al., 2020*). Molecular phylogenies estimate the genus of *Eleutherodactylus* diverged ~57 Mya (*Heinicke et al., 2007*; *Pyron, 2014*) and were present in the islands of the Greater Antilles by the mid-Cenozoic (~30 Mya; *Heinicke et al., 2007*). *E. coqui*, specifically, are estimated to have diverged from their nearest common ancestor ~7.4-12.8 Mya (*Pyron, 2014*; *Hedges et al., 2015*).

### Extra tympanic hearing

Vocal communication is central to the lives of most frogs that – like coquí – use sound as their central mode of communication. Effective communication, therefore, requires effective detection and localization of (conspecific) sounds. Sound localization in many species relies on neural comparisons of detection timing between ears, making two observations in diverse frogs very surprising. First, even species with very tiny heads, including *E. coqui*, are capable of accurate sound localization. Second, some frogs and toads perform sound localization in the complete absence of external ears (i.e. eardrums or tympanic membranes). These perplexing observations led to the hypothesis that there must be accessory, extra tympanic pathways for hearings. Studies in *E. coqui* were some of the first to solve this puzzle by confirming the existence of accessory hearing via the body wall and lungs (*Ehret et al., 1990*; *Narins et al., 1988*).

### Coquí in bioengineering

To make their impressive vocalizations, coquí frogs expand and contract their gular skin with remarkably high extensibility, elongating up to 400% for males (337% for females) with an ultimate tensile strength of 1.7 MPa, meaning the tissue is easily extensible. In comparison, the gular tissue of *Xenopus laevis*, which do not have vocal sacs, elongates up to 104% and requires much more force to extend with an ultimate tensile strength of ~6.3 MPa. This tissue has inspired research on the structure and molecular mechanisms of the coquí gular skin tissue compliance with implications for using biomimicry in development of more compliant biomaterials in regenerative medicine, such as 3D printed bladder tissues (*Hui et al., 2020*).

This altered cleavage pattern in turn shifts other developmental events (such as germ layer formation and neurulation) towards the animal pole (*Figure 5*; *Ninomiya et al., 2001*) and impacts the distribution of several early patterning RNAs (*Beckham et al., 2003*; *Fang et al., 2000*; *Fang and Elinson, 1999*).

Despite these differences, some early developmental features are conserved with metamorphosing frogs; germ cells are still located at the vegetal pole (*Figure 5*; *Elinson et al., 2011*) and neural crest cell migration is similar between *E. coqui* and metamorphosing frogs (*Moury and Hanken, 1995*). *Xenopus* is one of four model organisms (chick, mouse, zebrafish) that together have painted a picture of early vertebrate development in the past sixty years. Thus, coquí offer an important comparison for understanding the evolution of early vertebrate, and especially amphibian, development. Coquí also represents a natural 'experiment' in understanding how alterations of early molecular patterning impact later development.

The changes in early development and molecular organization of the coquí egg have prompted comparisons to the evolution of the amniote egg. The expansion of the vegetal region and shifting of the germ layers towards the animal pole is thought to also have occurred at the evolutionary transition from a membranous egg to the large amniote egg, a hypothesis supported by molecular data (*Arendt and Nübler-Jung, 1999*; *Buchholz et al., 2007*). Additionally, the covering of the yolk in *E. coqui* by dorsal structures, including muscles and skin, is a characteristic that may have been important in the evolution of amniotes (*Elinson and Fang, 1998*).

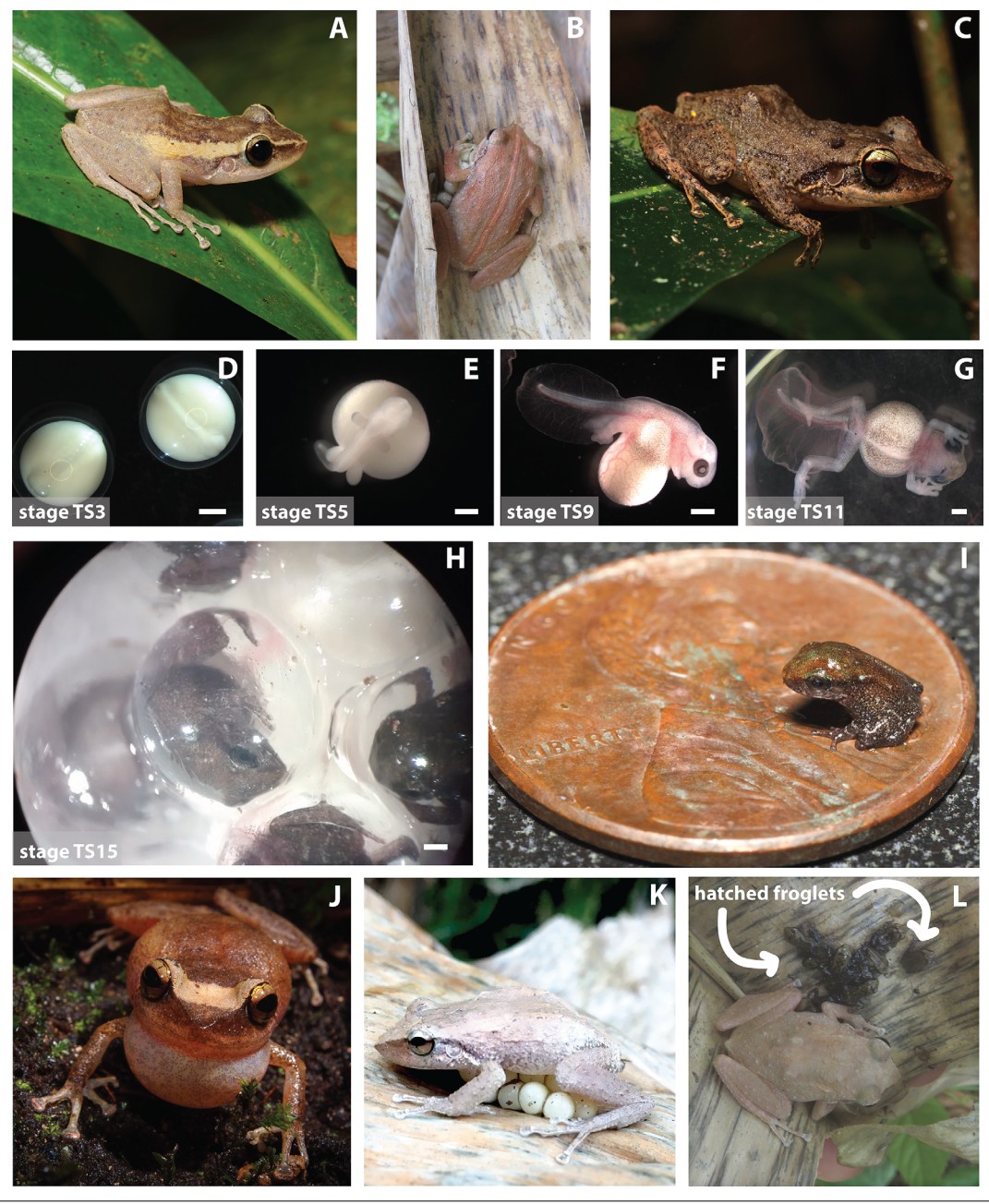

**Figure 2.** Different aspects of coquí natural history. (**A–C**) Coquí vary in color and pattern across different localities. (**D–H**) Coquí are direct developers, skipping the tadpole stage in their early development from an egg to a froglet. Townsend-Stewart (TS) described coquí development in 15 stages. Scale bars are 1 mm. (**I**) Froglets are quite small when they hatch and are at risk of predation by invertebrates. (**J**) Coquí are particularly known for their noisy calls made with their elastic vocal sacs. (**K–L**) Male coquí sit on their terrestrial eggs to hydrate them and will guard their newly hatched froglets.

Image credits: (**A, C**) S Van Belleghem (**B, K–L**) K Harmon (**D–H**) M Laslo (**I**) C Brown, USGS (**J**) A Lopez.

A final parallel between *E. coqui* and amniote development involves the nutritional endoderm. In the typical 1–2 mm amphibian egg, the whole egg becomes an embryo. In contrast, the nutritional endoderm is a novel coquí tissue made of cells that do not contribute to the embryonic intestine (as does regular endoderm) but do provide nutrition (*Buchholz et al., 2007*). This tissue has been proposed as an intermediate step in the evolution of the large amniote egg, as nutritional endoderm is like extraembryonic tissues in amniotes in that some oocyte

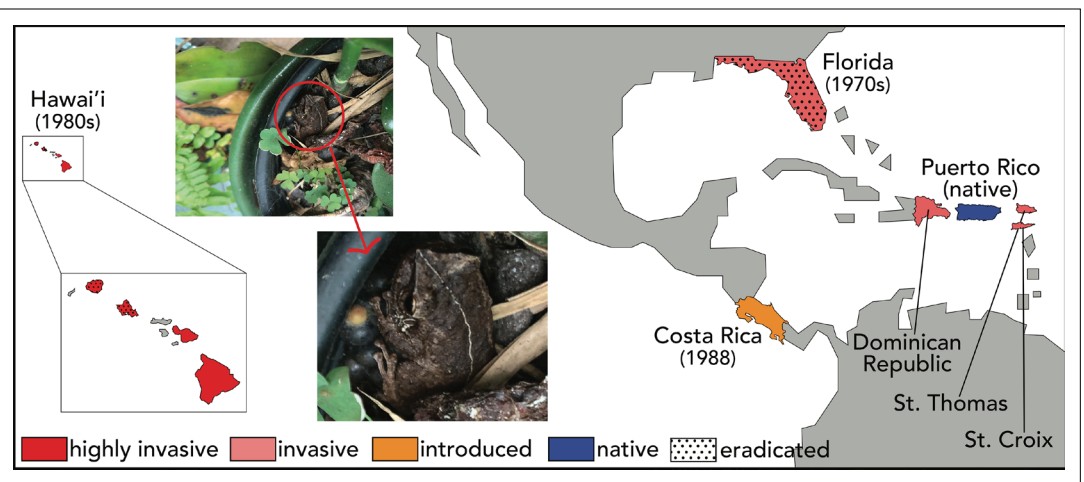

**Figure 3.** Distribution of the coquí. Native to Puerto Rico, *E. coqui* have successfully invaded many tropical islands and countries. Close to home, coquí have been introduced to other Caribbean islands, including the Culebra and Vieques Islands of Puerto Rico (***Rivero and Joglar, 1979***), St. Thomas and St. Croix in the Virgin Islands (***MacLean, 1982***), and the Dominican Republic (***Joglar, 1998***). Farther afield, coquí were accidentally introduced to Florida in the 1970s (***Austin and Schwartz, 1975***; ***Wilson and Porras, 1983***) and Hawai'i in the late 1980s (***Velo-Antón et al., 2007***), presumably as hitch-hikers on ornamental plants (photo inset). An intentional introduction of *E. coqui* was documented in Costa Rica, where six individuals were released in a private garden in 1998 and have since spread (***Barrantes-Madriga et al., 2019***). Floridian and some Hawaiian populations have been successfully eradicated due to inhospitable environmental conditions (Florida) and human effort (Hawai'i), but reintroductions remain a documented concern.

Image credit: RN Tischler.

material is used purely for nutrition and does not contribute to the embryo (***Elinson, 2009***). Therefore, coquí acts as a "missing link" or "transitional" model system to understand a possible evolutionary path between membranous eggs and the amniote egg.

### Late development

Post-embryonic development (metamorphosis) is shifted prior to hatching in coquí (into the period we call embryogenesis in coquí). Because coquí skip the tadpole stage, there are two potential trajectories of embryonic development: (1) larval structures never form, and adult features form directly, or (2) larval features develop and are remodeled into the adult morphology prior to hatching. Both of these cases are found for different structures in *E. coqui*.

Limbs are perhaps the most dramatic adult feature to form directly and early in coquí relative to metamorphosing frogs. Although the basic sequence and pattern of limb bone formation is conserved between *E. coqui* and metamorphosing frogs (***Hanken et al., 2001***), expression patterns of limb development genes differ, or are completely absent (***Gross et al., 2011***; ***Kerney and Hanken, 2008***). Another major difference in *E. coqui* limb development is the absence of

physical ridge of ectoderm (the AER, or apical ectodermal ridge; ***Richardson et al., 1998***) that promotes and maintains growth of the limb in all frogs and all other vertebrate models examined, with the exception of salamanders (***Hanken, 1986***; ***Saunders, 1998***; ***Saunders, 1948***; ***Tarin and Sturdee, 1971***). Although the physical structure is lacking in *E. coqui*, gene expression and transplantation studies suggest that major signaling centers are still present and function in coquí limbs as they do in other vertebrates (***Gross et al., 2011***; ***Hanken et al., 2001***).

Several other adult features appear directly in the adult configuration. During metamorphosis, adult muscle must be recruited from satellite cell populations, whereas in *E. coqui* the adult muscle fibers appear directly (***Hanken et al., 1997***). Adult nervous system features important for terrestrial life also form directly, including the olfactory system (***Jermakowicz et al., 2004***), the retina, and the optic tectum (***Schlosser and Roth, 1997***). In tadpoles, neurogenesis and proliferation of the spinal cord increase dramatically after hatching. These developmental events occur in the coquí spinal cord earlier relative to other parts of the central nervous system, likely because they are linked to early limb development (***Schlosser, 2003***).

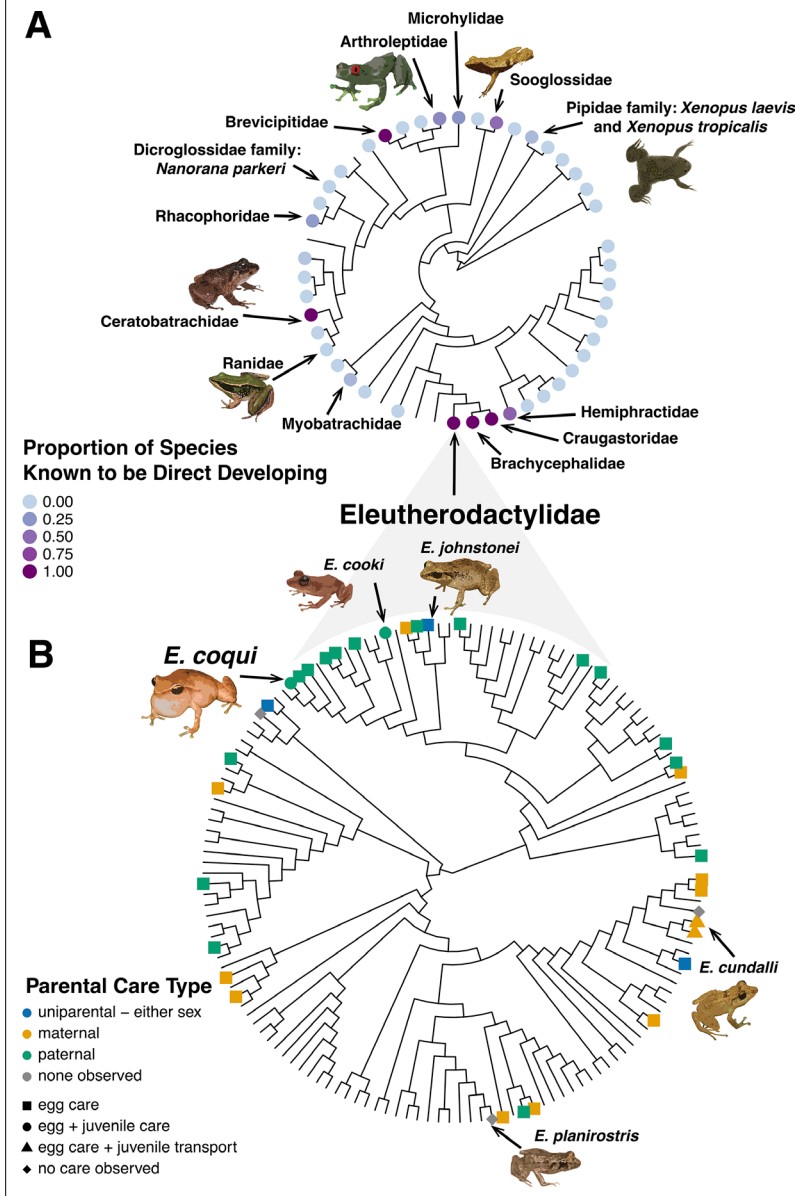

**Figure 4.** Evolution of direct development and parental care type. (**A**) Phylogenetic tree showing that, across Anura, there have been 11 independent evolutionary occurrences of direct development. Evidence suggests that the clade that includes Eleutherodactylidae, Craugastoridae, and Hemiphractidae is one of the oldest direct developing lineages, having evolved ~71–108 MYA (see *Heinicke et al., 2009* and *Gomez-Mestre et al., 2012* more details on the evolution of direct development). Other instances of direct development appear to have emerged more recently. (**B**) Phylogenetic tree showing that, across the family Eleutherodactylidae, there is variation in parental care strategies, including maternal, paternal, and amphisexual. *E. coqui* sits within a larger clade with mostly paternal egg care; additionally, male coquí care for froglets. *E. coqui* is one of two Eleutherodactylid species known to show paternal egg and juvenile care (*Furness and Capellini, 2019*). Currently, data is too sparse and varied to determine what specific parental care strategy was used by the ancestor of Eleutherodactylids; however, parental care is generally associated with direct development and a terrestrial life history (*Gomez-Mestre et al., 2012*; *Vági et al., 2019*). Phylogenetic clade relationships created from *Pyron, 2014* time tree data. Direct development and behavior traits mapped on phylogenies with data from *Furness and Capellini, 2019*.

Some larval features form briefly, and then are remodeled in the final third of development, which has therefore been nicknamed "cryptic" metamorphosis (*Callery and Elinson, 2000*). Many tadpole-specific cranial cartilages and muscles are patterned, or outlined, by gene expression, but the tissues never actually form (*Kerney et al., 2010*). The lower jaw and parts of the skeleton that would support the gills appear in the mid-metamorphic arrangement (*Hanken et al., 1992*; *Ziermann and Diogo, 2014*). Interestingly, some ancestral muscles important for aquatic feeding of the tadpole form appear briefly, although these muscles have no function in *E. coqui* (*Hanken et al., 1997*).

Finally, it is worth noting features of *E. coqui* development that are entirely novel and the typical tadpole-specific features that never form. The *E. coqui* tail is highly vascularized and may be used as a respiratory organ, a hypothesis supported by the observation that embryos undergoing accelerated development, including accelerated tail resorption, die if they cannot access air (ML, personal observation). The cement gland, which secretes a sticky mucus and allows tadpoles to secure themselves to substrate, and the lateral line, sensory cells that detect water movement and vibrations, never form (*Schlosser et al., 1999*). Altogether these observations inform our understanding of developmental modularity and constraint.

### Endocrine regulation of development

Given their unique life history, *E. coqui* are also useful for understanding the evolution of endocrine regulation of development. Thyroid hormone (TH) controls the timing of post-embryonic metamorphosis. Events normally under TH control in metamorphosing frogs appear to be a mixture of TH-independent and TH-dependent in *E. coqui*. For example, tail resorption and aspects of metabolism are regulated by TH as in metamorphing frogs (*Callery and Elinson, 2000*; *Elinson, 1994*), but limb development and growth, dependent on TH in metamorphosing frogs, seems to have both an early TH-independent and a late TH-dependent period in coquí (*Callery and Elinson, 2000*; *Elinson, 1994*). The hypothesis that maternally derived TH (*Elinson, 2013*; *Laslo et al., 2019*) could influence the early TH-independent period has not yet been directly tested. Other aspects of endocrine regulation of development appear to be conserved in *E. coqui* and metamorphosing frogs, including interactions

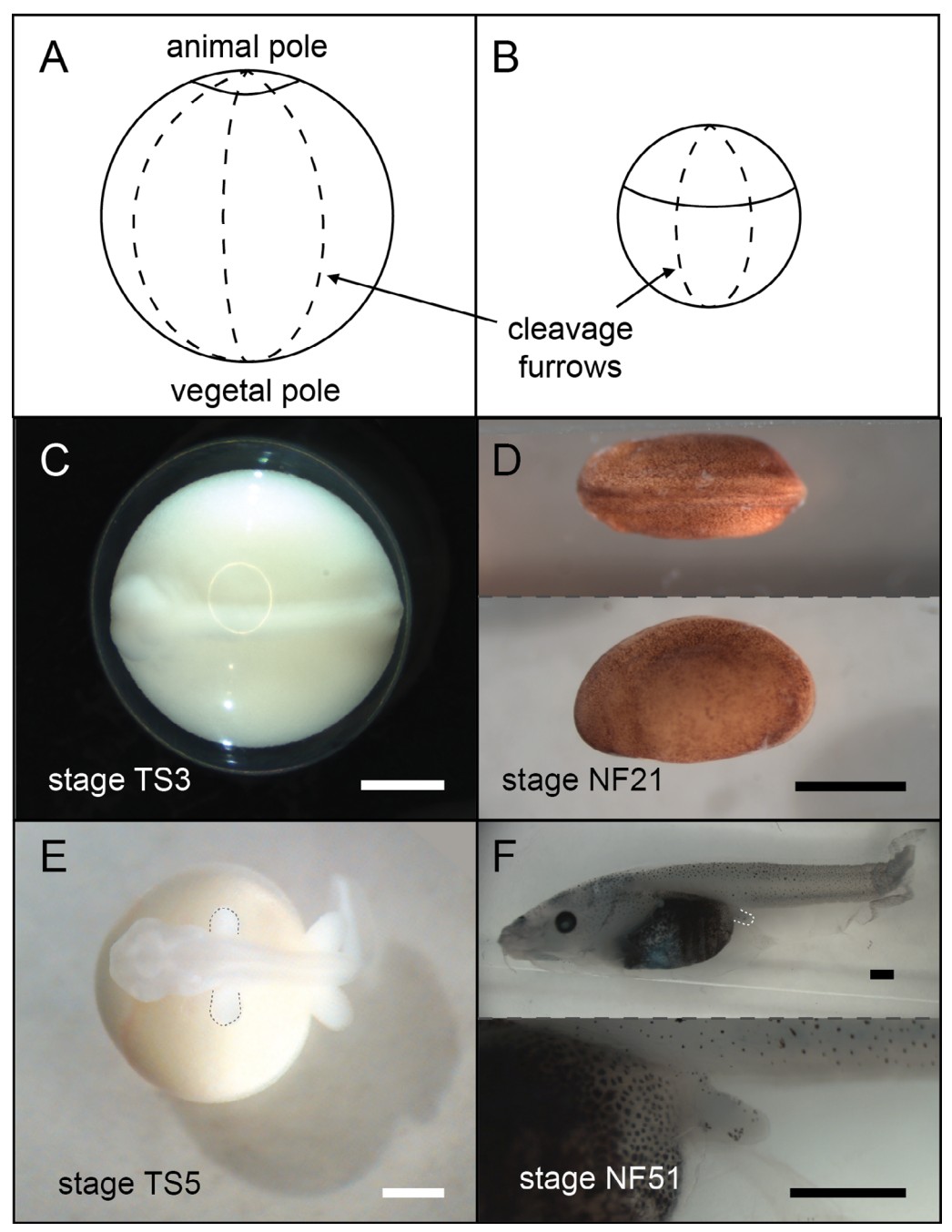

**Figure 5.** A comparison of *E.coqui* (**A, C, E**) and *X. laevis* (**B, D, F**) development. (**A–B**) Drawings of embryos after the first horizontal division. This horizontal division divides the embryo into animal (top) and vegetal (bottom) cells. The cap of *E. coqui* animal cells is much smaller than the animal cells in the early *X. laevis* embryo. (**C**) Dorsal view of the *E. coqui* neurula, which is shifted towards the animal pole compared to the *X. laevis* neurula (D; dorsal and lateral views shown). (**E–F**) Morphologically equivalent limb bud stages of the coquí embryo (**E**) and the *X. laevis* embryo (**F**). The coquí embryo is atop a large yolk mass, while limb buds emerge from a free-swimming *X. laevis* tadpole. Scale bars are 1 mm.

Image credit: M Laslo.

## Box 2. Outstanding questions about the natural history of *Eleutherodactylus coqui*

**Developmental biology**

- How does maternal nutritional and hormonal contribution impact early development and developmental timing (e.g., through yolk hormones)?

- How does heterochrony evolve and what are its genomic mechanisms (e.g., evolution of regulatory sequences or chromatin structure)?

- Do mechanisms mediating developmental plasticity in metamorphosing frogs also influence (plasticity in) developmental timing in coquí (e.g., glucocorticoids, thyroid hormone)?

- How does the mechanism of direct development in *E. coqui* compare to other direct-developing frogs?

**Neuroethology and social behavior**

- What behavioral, neural, and molecular mechanisms govern (switching between) distinct call types (aggressive vs courtship vs defense) in males (e.g., are distinct neural activity and/or hormonal patterns associated with distinct call types)?

- How are trade-offs between alternative behavioral states (territorial vs parental) negotiated at behavioral, physiological, and neural levels (e.g., do expression changes in shared or distinct gene sets mediate behavioral switching)?

- What are the neural and genomic molecular mechanisms mediating female mate choice of choice (e.g., what neural and hormonal changes are triggered by attractive male calls)?

**Conservation/invasion biology**

- How have invasive populations adapted to new environments despite low genetic divergence (e.g., in antipredator behavior or physiology)?

- How can ongoing introductions and new populations be used to track rapid adaptation to novel environments (e.g., expansions to high elevation sites in Hawai'i)?

- Have coquí adapted physiologically and/or behaviorally following eradication efforts (e.g., increased resistance to citric acid)?

**Integrative questions**

- *Proximate mechanisms of invasion*

  Are there genomic signatures of invasiveness? How does local adaptation take place in the face of low genetic diversity?
  How do natural history traits like developmental mode and communication play facilitate invasion and in turn evolve in response to invasion?
  How does the behavior of an invasive species contribute to ecosystem remodeling?
  What are the genetic and molecular mechanisms of Bd resistance in coquí and are they advantageous in the face of novel pathogens?

- *Parent-offspring interactions*

  What are the impacts of early life development and parental hormonal and nutritional contributions on later behavior in *E. coqui*?
  What are the maternal contributions to offspring in absence of direct parental care by females?

What is the role of signaling (chemical or hormonal) between adults and embryos/juveniles?
- *Comparative studies across Eleutherodactylids*

What can comparative genomic studies tell us about the development of alternative life history strategies?
How do neural mechanisms controlling behavior differ in species with different parental care strategies?
How does plasticity in development and behavior shape differences between species?

of TH and glucocorticoids, and the secretion of pituitary hormones by corticotropin-releasing hormone (*Jennings et al., 2015*; *Kulkarni et al., 2010*).

*E. coqui*'s direct-developing life history continues to teach us about the evolution of development. Coquí have taught us how egg size impacts early development, how the amniote egg may have evolved, how the vertebrate limb development module evolves, and how endocrine regulation of development evolves. These frogs also represent an exciting model for hormonal contributions to development. Thyroid hormone is necessary for neural development in vertebrates; however, this is difficult to study in mammals because thyroid hormones cross the placenta.

Coquí share many of the traits that make *Xenopus* species good models for translational research, including large and easily manipulated eggs; however, because coquí lack the extended free-swimming larval period of *Xenopus*, coquí's life cycle is more similar to the human development than *Xenopus*'. Additionally, evolutionary life cycle transitions across the animal tree of life (in fish, echinoderms, and insects) and within frogs can benefit from understanding the mechanism of direct development in coquí. Specifically, coquí is a valuable starting point for studies on the convergent evolution of direct development in frogs. Finally, previous studies have relied on a candidate gene approach. In the future, our understanding of development in *E. coqui* stands to benefit from modern sequencing technologies that allow an unbiased, exploratory approach to developmental questions. Because direct development is an evolutionary shift in developmental timing, and metamorphosis is relatively well understood, coquí represents an ideal model to investigate the proximate mechanisms, including the role of maternal hormones, in these shifts in developmental timing (*Box 2*).

## Social behavior

### *Vocal communication*

Vocal communication is central to the lives of most frogs. Among many possible choices, researchers were attracted to *E. coqui* due to their remarkably loud calls and sheer abundance. This initial choice was fortuitous, and studies of vocal communication in *E. coqui* have provided key insights.

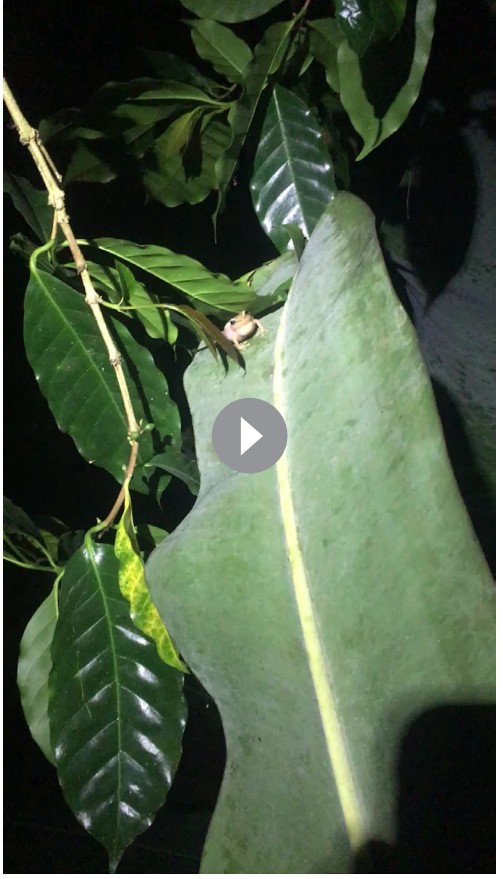

**Video 1.** Coquí are notorious in Puerto Rico for their loud vocalizations at night.

Video credit: K Harmon.

https://elifesciences.org/articles/73401/figures#video1

## Box 3. Natural diversity in *Eleutherodactylids*

*Eleutherodactylus* is a highly speciose genus ubiquitous across Neotropical regions (**Crawford and Smith, 2005**). Estimates for the number of species included in the genus *Eleutherodactylus* vary widely, from 185 to over 700; at the time of writing an AmphibiaWeb search for "Eleutherodactylus" returned 202 hits (**AmphibiaWeb, 2021**; **Duellman, 1993**; **Hedges et al., 2008**). While there was a high rate of newly described or resurrected taxa of Eleutherodactylids in the late 1990s and early 2000s (e.g., **Campbell and Savage, 2000**; **Duellman and Pramuk, 1999**; **Lynch, 2001**; **Savage and Myers, 2002**), further phylogenetic studies are needed to evaluate taxonomic relationships and biogeography of this diverse genus (**Crawford and Smith, 2005**; **Hedges et al., 2008**).

The life history diversity of Eleutherodactylids makes this genus a prime candidate for comparative studies and advancement as a powerful 'model clade' for integrative research (**Jourjine and Hoekstra, 2021**). Such work is embodied by research on *Peromyscus* mice (see **Bedford and Hoekstra, 2015**) and has potential in models like coquí and ricefish (see **Hilgers and Schwarzer, 2019**). For example, among *Eleutherodactylus* species there is broad variation in parental care strategies with species exhibiting uniparental care of eggs and/or froglets or no care (**Townsend, 1996**; **Figure 4B**). However, the sex providing uniparental care differs across species (**Townsend, 1996**; **Figure 4B**). *E. cooki* and *E. coqui* both show paternal care of eggs, whereas *E. cundalli* shows maternal care, and *E. johnstonei* shows amphisexual parental care of eggs (either paternal or maternal but not biparental; **Bourne, 1998**; **Burrowes, 2000**, p. 200; **Townsend, 1996**). Historically, *E. planirostris* was classified as exhibiting no parental care (**Townsend, 1996**) but recent observations suggest this species possibly has paternal care (**Iturriaga and Dugo-Cota, 2018**). Notably, *E. cundalli* mothers transport hatched froglets 'piggy-back' style (**Diesel et al., 1995**), as compared to the more common transportation of tadpoles (**Weygoldt, 2009**). Natural variation in Eleutherodactylidae provides an opportunity to understand the similarities and differences of these unique but analogous forms of parental care. In addition, most Eleutherodactylids are oviparous, but one species (*E. jasperi*) is ovoviviparous (**Drewry and Jones, 1976**), which opens the potential to investigate questions of development and maternal investment between closely related species. Finally, there is remarkable diversity in auditory tuning in this group (**Lewis et al., 1992**) providing fertile ground for comparative studies of multi-modal communication. The broad foundational knowledge about *E. coqui* provides an excellent starting point for integrative, comparative work.

Across Eleutherodactylidae, species vary in their International Union for Conservation of Nature (IUCN) Red List status. Unfortunately, like many amphibian genera (**Alford and Richards, 1999**), threats such as *Bd* and habitat destruction have decreased the population of most Eleutherodactylid species and many are considered endangered or critically endangered (**IUCN, 2021**). In contrast, *E. coqui* is one of only three known Eleutherodactylids that are increasing in population size (**IUCN, 2021**). The success of coquí as an invasive species may provide insights into why some amphibian species are thriving while others are struggling to maintain reproductive populations. Scientists across the Caribbean are working hard to conserve the biodiversity of this unique clade of frogs. For example, the non-profit organization El Proyecto Coquí (http://www.proyectocoqui.com/), led by Dr. Rafael L. Joglar at University of Puerto Rico at Ponce, works on conservation of amphibians and reptiles in Puerto Rico, with a particular focus on Eleutherodactylids, which, including the Puerto Rican coquí, are all often commonly referred to as coquí. Anyone interested in contributing to the conservation efforts of Puerto Rico can 'adopt' one of these beloved frogs through Conservación ConCiencia's 'Adopt-A-Coquí' program (https://www.conservacionconciencia.org/adopt-a-coqui).

*E. coqui* have a rich vocal repertoire, adjusting their namesake call for distinct social and environmental contexts. In the context of territory defense, males increase call rate and volume and vary the number of syllables in their call, sometimes using single 'co' notes and sometimes appending additional 'co' and 'qui' syllables (e.g., 'co-co-qui-qui', 'co-qui-qui-qui', etc.; *Narins and Capranica, 1976*; *Stewart and Rand, 1991*; see *Video 1*). Vocal escalation progresses to physical attack if an intruder comes within less 50cm (*Reyes-Campos, 1971*). By contrast, during courtship males reduce call volume and occasionally drop the 'qui' syllable (*Townsend and Stewart, 1986*; see below for additional details of courtship behavior).

Unlike in most frogs, female *E. coqui* also call. Female calls consist of one long note followed by a series of shorter notes that are of a distinct quality and quieter than male calls (*Stewart and Rand, 1991*). Females call to defend their daytime retreats and will physically attack intruders of both sexes (*Stewart and Rand, 1991*). In the wild coquí of any sex are found together in retreats only during courtship, and aggressive calls in both sexes likely allow frogs to appropriately space themselves and avoid costly physical encounters (*Stewart and Bishop, 1994*).

In addition to varying call volume, rate, and syllable number, the 'co' and 'qui' syllables themselves have distinct functions: the 'co' syllable is specialized for aggressive interactions while the 'qui' syllable is important for mate attraction (*Narins and Capranica, 1976*; *Narins and Capranica, 1978*). These functional specializations are associated with differences in auditory sensitivity between sexes (*Narins and Capranica, 1976*), a finding that was the first demonstration of sex differences in peripheral auditory tuning in a vertebrate.

Taken together these observations are exciting as they provide a mechanism by which males may simultaneously communicate specialized information to male competitors and female mates, even in a noisy, complex sound environment. More recent studies further highlight coquí as an intriguing example of how selection may fine-tune multi-functional communication systems by demonstrating behavioral and auditory tuning along an altitudinal gradient in *E. coqui* (*Meenderink et al., 2017*; *Meenderink et al., 2010*; *O'Neill and Beard, 2011*).

In brief, the complex interplay of note identity, loudness, and sequence alongside auditory tuning and behavioral adaptations remain fertile ground for investigation of complex, multi-functional vocal communication systems in *E. coqui*. Excitingly, genomic technologies for quantification and manipulation can now be brought to bear on open questions concerning the mechanisms of both signal production and reception.

## Courtship

Courtship in *E. coqui* is lengthy and involves a unique amplexus posture. Once a prospective mate arrives, the male leads the female to a nest site in his territory, moving 10-30cm ahead of the female and calling to encourage her to follow (*Townsend and Stewart, 1986*). This lead-and-follow continues until the pair reaches and enters a nest site. Once inside, the female inspects the site and – if she accepts – backs herself underneath the male (*Townsend, 1989*; *Townsend and Stewart, 1986*).

Unlike many amphibians, males do not clasp females. Instead, a female will lay her legs over the top of the male's legs in a "reverse hind leg clasp" (*Townsend and Stewart, 1986*). This unique posture is thought to facilitate internal fertilization, another coquí trait rare among frogs, with only 10 anuran species (0.13%) known to have internal fertilization (*Sever et al., 2003*).

Over the course of oviposition, the female moves out from under the male, such that he comes to rest on the newly laid clutch. While courtship generally commences in the early evening (peak calling in males is between dusk and midnight; *Woolbright, 1985*), oviposition does not occur until the next morning (*Townsend and Stewart, 1986*). Both the male and female generally remain in the nest until dusk of the following day (i.e. nearly 24 hours after courtship is initiated), at which time the male becomes aggressive, calling and biting to chase the female out of the nest. Data suggest that extended courtship is required for females to complete ovulation prior to oviposition (*Townsend and Stewart, 1986*). This research establishes great potential for studying connections between behavior and physiology in coquí.

## Parental care

Parental care is exhibited by only ~10% of anurans (*Wells, 2007*), making both the parental care and the direct development from embryo to froglet in *E. coqui* notable. Indeed, these unique aspects of coquí's life history may be linked because they eliminate the need for embryos and tadpoles to be in water. Reduced reliance on water for reproduction is favored because it reduces predation

risk to embryos and juveniles and allows frogs to take advantage of more varied habitats. Male coquí provide care to their offspring for 17–26 days until embryos hatch, often remaining with hatched froglets for an additional 1–6 days (*Townsend et al., 1984*). In the wild, males typically provide care to one clutch at a time and seldomly leave the nest during the brooding period (*Townsend, 1989*; *Townsend et al., 1984*). Care is critical for offspring survival and clutch failure is ~80% when males are removed due to desiccation, predation, and cannibalism (*Townsend et al., 1984*).

Parental care is likely energetically costly, both directly and indirectly, due to reduced opportunities for foraging, territory defense, and mating (*Townsend, 1989*). The role of cannibalism highlights these trade-offs. Coquí males cannibalize the eggs of conspecifics (simultaneous resource acquisition and aggression/competition) who vigorously defend their brood (an additional cost of care; *Townsend et al., 1984*). However, fathers are also known to cannibalize their own eggs, presumably to recover energetic resources when brood failure is likely due to high egg mortality from extrinsic factors (e.g., fungal infection), depredation, or cannibalism by another coquí (*Townsend et al., 1984*). As mechanistic links between parental behavior and infanticide have been demonstrated in mice (*Wu et al., 2014*) and suggested in other species (*Fischer and O'Connell, 2017*), these observations provide a particularly intriguing starting point for work exploring mechanisms of behavioral trade-offs.

Based on the mating and parental behavior detailed above, *E. coqui* males can be classified as being in a territorial versus a parental state. These behavioral states are largely mutually exclusive, begging the question of how alternative states are regulated. An early study showed that, as in many vertebrates, testosterone decreases during parental effort in *E. coqui* (*Townsend and Moger, 1987*).

More recent work has uncovered additional mechanisms mediating aggressive behavior, including a role for the nonapeptides arginine vasotocin and mesotocin (the amphibian homologs of mammalian vasopressin and oxytocin, respectively), as well as serotonin. Increased vasotocin or oxytocin increased aggressive calling behavior and the propensity for non-territorial satellite males to establish a new territory (*Ten Eyck and Ten Eyck, 2020*; *Ten Eyck and Ten Eyck, 2017*; *Ten Eyck and ul Haq, 2012*), while increased serotonin signaling decreased calling

in male territory holders (*Ten Eyck, 2008*; *Ten Eyck and Ten Eyck, 2020*).

Interestingly, vasotocin treatment did not increase calling in parental (non-calling, non-territorial) males, suggesting that the effects of vasotocin treatment are modulated by parental state (*Ten Eyck and ul Haq, 2012*). Nonapeptides are broadly implicated in vertebrate sociality including parental care and aggression (*Goodson, 2013*; *Goodson, 2008*), and *E. coqui* provide opportunities to understand how these molecules mediate social behavioral trade-offs not only among species but also at the individual level. Given early indications that parental state influences the effects of neuromodulators on calling behavior, it will be interesting to explore the effects of these molecules on the different call types discussed above, which are produced for distinct purposes in different behavioral contexts.

In sum, coquí social behavior provides opportunities for exploring sensory processing, behavioral trade-offs, and how and why unique life histories evolve. Existing studies address various aspects of these questions and set the stage for integration across behavioral contexts and levels of analysis. For example, previous studies have characterized differences in auditory tuning in males versus females (*Narins and Capranica, 1976*) and the molecular correlates of calling in males (*Ten Eyck, 2008*; *Ten Eyck and ul Haq, 2012*), and it would now be fruitful to disentangle neural mechanisms of distinct call types, to explore neural and molecular mechanisms of vocalization in females, and to ask whether the same neuromodulators that influence calling behavior also modulate auditory sensitivity within and between sexes (see *Box 2*).

## Biological invasion

While coquí are a beloved cultural symbol in their native Puerto Rico, they are much less celebrated – and even despised – where they have invaded in Central America, North America, and the Caribbean (*Figure 3*). With the notable exception of cane toads, *Rhinella marina*, in Australia (see *Shine, 2010* for review), invasive amphibians are relatively rare. The generalist lifestyle of *E. coqui* and their independence from standing water for reproduction have contributed to coquí's rapid dispersal and proliferation, as have both accidental and intentional human transport (*Everman et al., 2013*; *Kraus and Campbell III, 2002*).

*E. coqui* are considered among the 100 worst invasive species in the world (*Simberloff and Rejmanek, 2019*), bringing with them a host of

biological and economic problems. The Hawaiian invasion is both the most extensive and the best studied. Hawaiian populations reach densities of up to 91,000 frogs/hectare (*Beard et al., 2009*), two to three times higher than estimates for native populations in Puerto Rico (*Woolbright et al., 2006*), and among the highest known for any amphibian worldwide. From a biological perspective, the invasion of coquí presents several concerns, most notably competition with and predation on endemic species (*Beard, 2007*; *Choi and Beard, 2011*). From an economic perspective, these small frogs are responsible for millions of dollars of damage each year (*Kaiser and Burnett, 2006*), decreasing property values (chorusing coquí produce sound pressures of up to 95 dB – similar to a motorcycle engine) and impacting multi-million-dollar floriculture industry (*Beard and Pitt, 2005*). Nonetheless, having coquí on their property decreases residents' negative perceptions (*Kalnicky et al., 2014*) suggesting attitudes toward these noisy invaders may shift over time.

In view of biological and economic concerns, substantial resources have been invested in eradicating *E. coqui* in Hawai'i (reviewed *Beard and Pitt, 2012*). While eradication is likely no longer possible on the Big Island, smaller populations on O'ahu and Kauai were successfully eradicated (*Beachy et al., 2011*), although reintroductions remain a constant concern. Hot water treatment is effective in greenhouse settings and treatment with citric acid is approved by the USDA. The chytrid fungus *Batrachochytrium dendrobatidis* (Bd) was proposed as a biological control agent, but this idea has been largely abandoned as coquí are somewhat resistant to the fungus (*Langhammer et al., 2014*; *Rollins-Smith et al., 2015*) and due to concerns that Bd could spread to vulnerable amphibians outside Hawai'i (*Beard and O'Neill, 2005*). Notably, the above methods are possible in Hawai'i because the archipelago does not have any native amphibians, but are unlikely to translate well to other coquí invasions – for example in Costa Rica – where protection of endemic species is of central concern.

While the invasion of coquí presents serious biological and economic concerns, it also presents opportunities for basic and applied research (see *Hanson et al., 2020* for a similar example in house sparrows, and the excellent body of work on cane toad invasion in Australia; e.g., *Brown et al., 2015*; *DeVore et al., 2021*; *Rollins et al., 2015*; *Selechnik et al., 2017*). Conservation-centered questions abound and excellent research is being conducted in this area (*Beard*

*et al., 2009*; *Beard and Pitt, 2005*; *Joglar, 1998*). Additionally, comparisons between invasive and native Puerto Rican populations provide a largely untapped opportunity for exploration of local adaptation and examination of genomic and phenotypic traits that facilitate invasion. This research could shed light on why *E. coqui* have been so successful – both in their native and introduced ranges – while some closely related species are imperiled. *E. coqui*'s relative resistance to Bd (*Langhammer et al., 2014*; *Rollins-Smith et al., 2015*), which is driving many amphibian species to extinction, may provide particularly important insights. Finally, large, invasive populations allow for developmental and neurobiological studies that often require destructive sampling and would therefore not be ethical in most wild populations. Alongside explicit consideration of their natural history, study of model organisms outside the lab has the potential to provide new insights through the incorporation of critical environmental and ecological context.

## Conclusion

Thanks to their unique natural history, coquí have attracted the interest of diverse research fields, including some outside the scope of this paper (*Box 1*).

Direct development from egg to froglet has made coquí an excellent model system for evolutionary developmental biologists to examine how changes in maternal provisioning influence development and speculate on the evolution of the amniotic egg. Studies of coquí development have uncovered new evolutionary features and probed how coquí have altered conserved developmental processes in the transition from a biphasic life history to direct development. In turn, direct development has facilitated the coquí's invasion potential, making these frogs a focus of conservation research. Through direct development and parental care, coquí have reduced reliance on water which opens many ecological niches in comparison to water-bound amphibians and facilitated the invasion of coquí on many tropical islands. Coquí's impressively loud nocturnal choruses, in and outside their native range, have also attracted researchers in behavioral neuroscience, yielding work that contributes to our understanding of auditory communication and adaptations for sensory processing.

Despite this rich history of research, there are still unexplored aspects of coquí natural history that are driving novel research. For instance, coquí have recently shown promise in advancing

bioengineering due to incredibly extensibile gular skin tissue that allows them to produce their characteristic vocalizations (*Hui et al., 2020*; *Box 1*).

Despite great interest in the natural history of coquí across diverse fields, there is little work bridging disparate disciplines. The time is ripe to leverage previous studies and emerging technologies to build a holistic understanding of this culturally, developmentally, behaviorally, and evolutionarily interesting species. In addition to discipline-spanning work, coquí are poised for integrative research spanning levels of biological organization, from molecular mechanisms to ecosystem impacts. Recent and foundational studies provide the necessary framework for impactful application of genomic, transcriptomic, and genetic manipulation techniques: transcriptomic approaches have already been successfully applied (*Laslo, 2019*), a genome is currently under construction (Vert Genome Project, https://vertebrategenomesproject.org/), and tools for genomic manipulation are particularly amenable in amphibians, which have large, externally developing embryos.

The intersection of traditionally distinct conservation, developmental, and neuroethological work provides fertile ground for impactful integrative studies; for example, linking developmental manipulations with adult behavioral outcomes, understanding the evolution and development of alternative life history strategies, understanding the impacts of maternal hormonal and nutritional contributions to offspring health and developmental timing in the absence of direct care, and exploring the role of developmental mode and communication in invasion potential, to name just a few (see *Box 2*). Moreover, diversity across closely related Eleutherodactylids provides opportunities to expand these questions into a powerful comparative framework (*Jourjine and Hoekstra, 2021*; see *Box 3*).

Beyond the science, *E. coqui* are valuable and unique in their cultural importance. The cultural interest in this small, vocal frog provides an opportunity to engage diverse communities in STEM educational opportunities and citizen science efforts. Coquí are also well-positioned to welcome and include diverse stakeholders in scientific practice.

Given the geographic range of coquí, research involving these frogs can work towards the decolonization of scientific practice by partnering with local researchers and community members, particularly in Puerto Rico and Hawai'i. These partnerships are mutually beneficial as local scientific and lay communities have a unique knowledge of and relationship with their local flora and fauna but often lack the infrastructure and training opportunities that promote certain subdisciplines (for example, ready access to expensive, rapidly evolving genomic technologies).

Meaningful collaboration can improve the science by increasing diversity of experience and thought, and help increase opportunities where they are lacking. Collaborating with local scientists on coquí research in its native and invasive range will help increase the representation of Puerto Rican and other Latinx countries in ecology, neuroscience, and conservation. Coquí provide an opportunity for Puerto Rican scientists to do high-quality work close to home and in fields historically lacking Latinx representation (*Lewis et al., 2009*; *O'Brien et al., 2020*; *Quirk, 2017*; *Rincón and Rodriguez, 2020*). It is difficult to quantify the representation of Native Hawaiian and Pacific Islanders in STEM because Pacific Islanders are traditionally grouped together with Asian Americans, despite being a group with multiple racial and ethnic identities (*Maramba, 2013*). Participation is likely lower than Latinx (*Levine, 2013*; *Maramba, 2013*; *NSF and NCSES, 2021*), and the invasion of coquí in Hawai'i provides an opportunity for community engagement and for training Hawaiian scientists in their homelands on research questions with direct local application.

We hope that the broad base of scientific and cultural knowledge about the unique life history of *E. coqui* will inspire researchers to capitalize on the big potential of these small frogs to do integrative work in the lab, the field, and the community.

## Acknowledgements
We thank members of the Fischer Lab and Mark Hauber for their helpful and encouraging feedback. We also thank Ryan Kerney and two anonymous reviewers for their insights and constructive feedback.

**Sarah E Westrick** is in the Department of Evolution, Ecology, and Behavior, University of Illinois Urbana-Champaign, Urbana, United States
westse@illinois.edu
http://orcid.org/0000-0002-5381-1048
**Mara Laslo** is in the Curriculum Fellow Program, Harvard Medical School, Harvard University, Cambridge, United States
http://orcid.org/0000-0003-4022-4327

Eva K Fischer is in the Department of Evolution, Ecology and Behavior, University of Illinois Urbana-Champaign, Urbana, United States

http://orcid.org/0000-0002-2916-0900

*Author contributions:* Sarah E Westrick, Conceptualization, Visualization, Writing – original draft, Writing – review and editing; Mara Laslo, Conceptualization, Visualization, Writing – original draft, Writing – review and editing; Eva K Fischer, Conceptualization, Visualization, Writing – original draft, Writing – review and editing

*Competing interests:* The authors declare that no competing interests exist.

## Funding

| Funder | Grant reference number | Author |
|---|---|---|
| National Science Foundation | Postdoctoral Fellowship in Biology | Sarah E Westrick |
| Hanse-Wissenschaftskolleg Institute for Advanced Study | | Eva K Fischer |

The funders had no role in study design, data collection and interpretation, or the decision to submit the work for publication.

## Decision letter and Author response
Decision letter https://doi.org/10.7554/eLife.73401.sa1
Author response https://doi.org/10.7554/eLife.73401.sa2

# Additional files

## Supplementary files
• Transparent reporting form

## Data availability
No new data were generated for this article. Data from Furness and Capellini (2019) and from Pyron (2014; deposited in Dryad: https://doi.org/10.5061/dryad.jm453) were used.

The following previously published dataset was used:

| Author(s) | Year | Dataset URL | Database and Identifier |
|---|---|---|---|
| Pyron AR | 2014 | https://doi.org/10.5061/dryad.jm453 | Dryad Digital Repository, 10.5061/dryad.jm453 |

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
