## [Decision Letter]

**Decision letter after peer review:**

Thank you for submitting your article "The big potential of the small frog Eleutherodactylus coqui " to *eLife* for consideration as a Feature Article. Your article has been reviewed by 3 peer reviewers, and the evaluation has been overseen by a member of the *eLife* Features Team (Helena Pérez Valle). The following individual involved in review of your submission has agreed to reveal their identity: Ryan Kerney.

The reviewers and editors have discussed the reviews and we have drafted this decision letter to help you prepare a revised submission.

Summary:

This is a useful synthesis from multiple perspectives on the research surrounding the frog Eleutherodactylus coqui. Like so many focal species in biological research, there are non-overlapping research programs from a range of disciplines that are converging on this fascinating frog. This review will be of interest to a wide range of specialists, since the article provides not only a very broad overview of the current knowledge about the biology and life history of E. coqui, but also points out potential major directions for future research.

Essential revisions:

1. Please update Figure 4 to include more information about the convergence of direct development in amphibians and parental care in Eleutherodactylids. Some families with direct development are missing from the tree labels (e.g. Rhacophoridae) and other clades with direct development on the tree are not labelled (e.g. within the clade that includes Arthroleptidae). See Kentwood Wells (Wells K. 2007. The Ecology and Behavior of Amphibians. Chicago: The University of Chicago Press) to help clarify 4A. Please include (if possible) when direct development or parental types evolved, especially in relation to coquí frogs: e.g. are coquís among the oldest lineages to have direct development and parental care? Or are there other related species that developed it first? Please also provide more information about the paternal/maternal care species named in 4B. Please also point out that coqui offers a basis of comparison for studies on the convergence of direct development, which evolved independently in multiple amphibian lineages. The work by Rick Shine on cane toad physiology and evolution in Australia may provide another interesting basis of comparison to the coqui invasions. See:

– Rollins LA, Richardson MF, Shine R. A genetic perspective on rapid evolution in cane toads (Rhinella marina). Mol Ecol. 2015 May;24(9):2264-76. doi: 10.1111/mec.13184. Epub 2015 Apr 20. PMID: 25894012

– Selechnik D, Rollins LA, Brown GP, Kelehear C, Shine R. The things they carried: The pathogenic effects of old and new parasites following the intercontinental invasion of the Australian cane toad (*Rhinella marina*). Int J Parasitol Parasites Wildl. 2016 Dec 29;6(3):375-385. doi: 10.1016/j.ijppaw.2016.12.001. PMID: 30951567; PMCID: PMC5715224

– Shine R. The ecological impact of invasive cane toads (Bufo marinus) in Australia. Q Rev Biol. 2010 Sep;85(3):253-91. doi: 10.1086/655116. PMID: 20919631

2. In Figure 3, please confirm that coquís are currently present in Florida, given that they were thought to disappear during a cold winter around the year 2,000. They are also no longer on Kauai and Oahu since they were eradicated, so please remove those from the map.

3. In box 2 please ensure that the questions are specific and testable. For example:

– The question about developmental plasticity in juveniles is vague, please add more details as to what developmental plasticity means in the context of the coquí, and what traits would be studied.

– The question regarding the role of female mate choice seems quite broad, but it is listed under neuroethology and social behaviour. Would it be possible to rephrase the question to frame it more specifically within this heading?

– There is already some research into how native and invasive populations have diverged, as well as some investigation into rapid adaptation in Hawaii. See work by Eric O'Neill: O'Neill, EM, KH Beard, CW Fox. 2018. Body size and life history traits in native and introduced population of Coqui frogs. Copeia; O'Neill, EM, KH Beard, CW Fox. 2012. Cast adrift on an island: introduced populations experience an altered balance between selection and drift. Biology Letters.

– How do the mechanisms of Bd resistance relate to invasion per se?

---

## [Author Response]

Essential revisions:1. Please update Figure 4 to include more information about the convergence of direct development in amphibians and parental care in Eleutherodactylids. Some families with direct development are missing from the tree labels (e.g. Rhacophoridae) and other clades with direct development on the tree are not labelled (e.g. within the clade that includes Arthroleptidae). See Kentwood Wells (Wells K. 2007. The Ecology and Behavior of Amphibians. Chicago: The University of Chicago Press) to help clarify 4A.

We have labeled the remaining clades containing direct developing species using data from Furness and Capellini (2019). We also corrected an error in the original figure in the location of Pipidae and have corrected it in the revision.

Please include (if possible) when direct development or parental types evolved, especially in relation to coquí frogs: e.g. are coquís among the oldest lineages to have direct development and parental care? Or are there other related species that developed it first?

We have updated the figure caption for Figure 4 to include details on when direct development in Eleutherodactylids is likely to have evolved relative to other independent occurrences of direct development. Because data are sparse for Eleutherodactylids, we do not know what the ancestral parental care type for Eleutherodactylids is but it is likely to have exhibited some type of parental care given the terrestrial nature and direct development of Eleutherodactylids.

Please also provide more information about the paternal/maternal care species named in 4B.

We have added more details about the labeled Eleutherodactylus species in Box 3. We chose these species to showcase the diversity in parental care strategies in this genus.

Please also point out that coqui offers a basis of comparison for studies on the convergence of direct development, which evolved independently in multiple amphibian lineages.

We have added a sentence describing this point at the end of “Development” section, as well as a question in Box 2*.*

The work by Rick Shine on cane toad physiology and evolution in Australia may provide another interesting basis of comparison to the coqui invasions. See:– Rollins LA, Richardson MF, Shine R. A genetic perspective on rapid evolution in cane toads (Rhinella marina). Mol Ecol. 2015 May;24(9):2264-76. doi: 10.1111/mec.13184. Epub 2015 Apr 20. PMID: 25894012– Selechnik D, Rollins LA, Brown GP, Kelehear C, Shine R. The things they carried: The pathogenic effects of old and new parasites following the intercontinental invasion of the Australian cane toad (Rhinella marina). Int J Parasitol Parasites Wildl. 2016 Dec 29;6(3):375-385. doi: 10.1016/j.ijppaw.2016.12.001. PMID: 30951567; PMCID: PMC5715224– Shine R. The ecological impact of invasive cane toads (Bufo marinus) in Australia. Q Rev Biol. 2010 Sep;85(3):253-91. doi: 10.1086/655116. PMID: 20919631

We agree and have modified the next to include references to the excellent body of work on the invasion of cane toads in Australia.

2. In Figure 3, please confirm that coquís are currently present in Florida, given that they were thought to disappear during a cold winter around the year 2,000. They are also no longer on Kauai and Oahu since they were eradicated, so please remove those from the map.

We have modified the figure and figure legend to indicate invasive populations in Florida and Hawai’i that have been eradicated.

3. In box 2 please ensure that the questions are specific and testable. For example:

As in other articles in the series, we have intentionally left these questions quite broad as there are many possible avenues for exciting future research. Nonetheless, we have refined our language and included more detailed examples.

– The question about developmental plasticity in juveniles is vague, please add more details as to what developmental plasticity means in the context of the coquí, and what traits would be studied.

Done

– The question regarding the role of female mate choice seems quite broad, but it is listed under neuroethology and social behaviour. Would it be possible to rephrase the question to frame it more specifically within this heading?

As work on call processing has been done we have modified this point to suggest that examining neural and hormonal shifts in response to an attractive call is a promising starting point. As little is known about female choice in this system a wealth of exciting experiments are possible.

– There is already some research into how native and invasive populations have diverged, as well as some investigation into rapid adaptation in Hawaii. See work by Eric O'Neill: O'Neill, EM, KH Beard, CW Fox. 2018. Body size and life history traits in native and introduced population of Coqui frogs. Copeia; O'Neill, EM, KH Beard, CW Fox. 2012. Cast adrift on an island: introduced populations experience an altered balance between selection and drift. Biology Letters.

We agree these studies are interesting and have revised our wording to clarify that we therefore advocate for additional studies in this area.

– How do the mechanisms of Bd resistance relate to invasion per se?

We have moved this point to the ‘Integrative Questions’ as we feel it fits better there.